# A Proposal to Differentiate ACO, Asthma and COPD in Vietnam

**DOI:** 10.3390/jpm13010078

**Published:** 2022-12-29

**Authors:** Ha Thi Chu, Thuy Chau Nguyen, Isabelle Godin, Olivier Michel

**Affiliations:** 1Outpatient Department, Pham Ngoc Thach Hospital, Ho Chi Minh City 70000, Vietnam; 2Polyclinic, Department of Family Medicine, Pham Ngoc Thach University of Medicine, Ho Chi Minh City 70000, Vietnam; 3School of Public Health, Université Libre de Bruxelles (ULB), 1050 Bruxelles, Belgium; 4Clinic of Immuno-Allergology, CHU Brugmann (ULB), 1050 Bruxelles, Belgium

**Keywords:** asthma, COPD, overlap, Vietnam, spirometry, cumulative smoking, diffusion capacity

## Abstract

Background: In low- and middle-income countries, such as Vietnam, the population is exposed to multiple risks, leading to frequent allergic asthma, COPD and their overlap (ACO). We aimed to differentiate asthma and COPD, so that recommended treatments can be applied. Methods: We hypothesized that during life, the cumulative exposure to noxious particles increases the relative prevalence of COPD, while due to immuno-senescence, the prevalence of allergic asthma decreases with age. Among 568 patients with chronic respiratory symptoms, five phenotypes were defined, based on responsiveness to a bronchodilator (BD), diffusion capacity and cumulative smoking. Then the relative prevalence of each phenotype was related with age. Results: the smoker BD irreversible patients were considered “COPD”, while the full BD responders and non-smoking BD incomplete responders were “asthmatics”. The other patients were ACO, distributed as “like-COPD” or “like-asthma”, based on decreased or normal diffusion capacity. The relative prevalence of asthma, COPD and ACO were 26, 42 and 32% (18% “like-asthma”, 14% “like-COPD”). Conclusion: Vietnamese patients with chronic respiratory symptoms were considered as falling into asthma or COPD groups, based on cumulative smoking, spirometry with reversibility and diffusion capacity. The relative prevalence of asthma and COPD were 44 and 56%, respectively, most of which did not require corticosteroids.

## 1. Introduction

The more frequent Chronic Obstructive Respiratory Diseases (CORDs) are asthma, COPD and their overlap (ACO) [1,2,3,4]. In low- and middle-income countries (LMICs), characterized by rapid economic growth, exposure to CORD risk factors is different compared to Western countries. For example, in Vietnam, the rural population, accounting for about 65% of the total population, is frequently infested by chronic parasitosis and exposed to fumes from incense burning, biomass fuel and organic pro-inflammatory dusts [5]. The urban population is exposed to high traffic-related air pollution and poorly ventilated dwellings, associated with a higher risk of dust mite sensitization (at least partially consequent to the control of parasitosis) [6], and to noxious occupational fumes and particles. About 50% of Vietnamese men are heavy smokers [7]. The intense rural to urban migration reinforces the risk of developing allergies [6]. Besides environmental factors, sequelae of pulmonary tuberculosis are frequent and genetic factors participate to individual susceptibility to noxious airborne agents, both contributing to the risk for COPD [8,9,10]. Thus, the Vietnamese population is exposed to multiple risks of both allergic asthma and COPD and consequently, a high prevalence of ACO can be assumed [11].

Oral and/or inhaled corticosteroids (ICSs) are essential in asthmatics to reduce the risk of severe exacerbations and death, and, among COPD sufferers, long-acting beta-mimetics and/or muscarinic antagonists are recommended as the initial treatment [12]. In Vietnam, most patients with CORD are treated with (ICS), whatever the specific diagnosis. The consequence is costly overtreatment with an increased risk of pneumonia [13,14]. and tuberculosis (TB), as reported in intermediate TB-burden countries [15]. On one side, to limit this over-cost and the side effects, there is a need to apply the Global Initiative for Asthma (GINA) and Global Initiative for Chronic Obstructive Lung Disease (GOLD) guidelines of treatments [12,16]. On the other hand, in Vietnam, the diagnosis of asthma or COPD [1,2,3,4] is difficult, due to the wide range of risk factors with frequent ACOs.

The aim of this study was to develop a simplified strategy to differentiate asthma, ACO and COPD in Vietnam.

## 2. Materials and Methods

### 2.1. Population

Patients with chronic respiratory symptoms (cough and/or sputum and/or dyspnea and/or wheezing and/or chest tightness for > 3 months) were prospectively preselected, after exclusion of TB at Pham Ngoc Thach hospital, the reference for respiratory diseases in Ho Chi Minh City, as reported previously [6]. In brief, from 2014 to 2015, of the 1000 daily outpatients reporting respiratory symptoms, about 700 patients were excluded, due to active tuberculosis, cancer and acute infectious diseases. Patients with significant lung defects (*n* = 120) were invited to participate in the study by random sampling (1/10), and, after providing written informed consent, 5–6 patients were included daily, to give a total of 610 patients. Among these, a total of 568 CORD were analyzed in this study. Participants were distributed into 5 age groups, corresponding to a balanced distribution (i.e., 20–43, 44–53, 53–59, 60–65, 66–90 years).

The study was approved by the Ethics Committee of Pham Ngoc Thach hospital (CS/PT/13/12) and each patient gave informed written consent.

### 2.2. Questionnaire

Clinical and environmental characteristics of the studied population were obtained from the questionnaire, reported previously [6]. In brief, this questionnaire was adapted for the Vietnamese population from validated questionnaires for the evaluation of patients with chronic respiratory diseases and indoor air pollution (European Community Respiratory Health Survey-1 and Health survey-2). The questionnaire contained questions on demographic information, medical history, smoking habits, occupational history, type of dwelling and details about exposure to indoor pollution.

Exposure to indoor fumes was defined as a patient exposed for at least 10 years to fumes from biomass and/or incense.

### 2.3. Lung Function and Allergy Evaluation

The lung function tests were measured with a Master Screen PFT (Carefusion Ltd., Hochberg, Germany), according to the recommendations of the American Thoracic Society/European Respiratory Society (ERS) [17], by a trained technician [18]. Diffusing capacity of the Lungs for Carbon Monoxide (DLCO) was obtained by the gas dilution method. Compared to baseline values, an increase of >200 mL and >12% in forced expiratory volume in 1 s (FEV_1_) after the bronchodilator (BD) [400 mcg salbutamol], was considered significant. In parallel, the new ATS/ERS technical standard was also used to define significant bronchial reversibility, as a change of >10% FEV1 predicted value [19]. Predicted values were referenced from the values of the East Asian population, i.e., the ERS reference values with a 0.9 correction factor [20].

A skin prick test (SPT) (*Dermatophagoides pteronyssinus* and *farinae* and *Blomia tropicalis*) was considered positive for a wheal reaction ≥ 4 mm compared to the negative control. The concentrations of IgE to Dpt, Blo t extract, and the recombinants proteins Der p1, Der p2, Der p23 were measured by ImmunoCAP (Phadia IDM 1000, Uppsala, Sweden).

### 2.4. Spirometric Definitions (Figure 1)

The definitions of airways obstruction and reversibility were those of the GINA and GOLD guidelines [15,16].

An “obstructive airway disease” was defined by chronic respiratory symptoms lasting > 3 months and baseline FEV_1_/Forced Vital Capacity (FVC) < Lower Limit of Normal (LLN) of predicted value (PV). Compared to baseline values, an increase of >200 mL and >12% FEV1 after BD [400 mcg salbutamol], was considered as significant.

**Figure 1 jpm-13-00078-f001:**
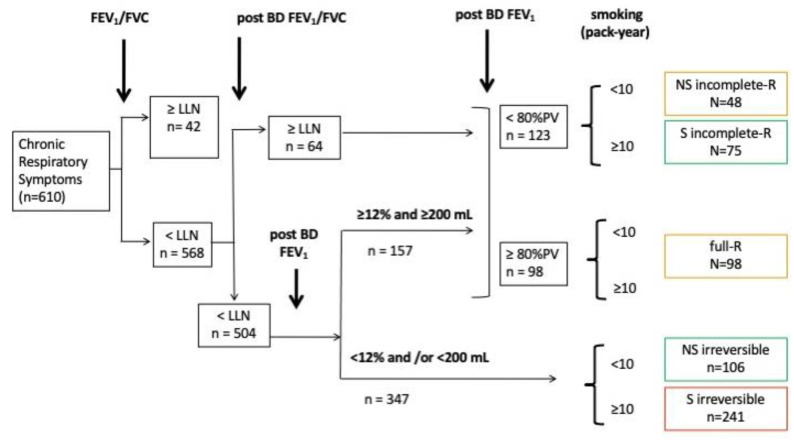
Definition and prevalence of 5 phenotypes of obstructive airway diseases. FEV_1_: forced expiratory volume in 1 s; FVC: forced vital capacity. BD: bronchodilator; COPD: chronic obstructive pulmonary disease. NS: Non-Smoker. S: Smoker. PV: Predicted Value. -R: Responder.

The BD response of both FEV_1_/FVC and FEV_1_ to salbutamol (400 mcg) inhalation was defined as:a “responder” was a post-BD FEV_1_/FVC ≥ LLN or an increase of FEV1 ≥ 200 mL and ≥ 12%. Among them:○a “full responder” was a “responder” with a post BD FEV_1_ > 80% PV.○an “incomplete responder” was a “responder” with a post BD FEV_1_ < 80% PV.an “irreversible” was a patient with a post BD response inducing a FEV_1_/FVC ≤ LLN and no significant increase of FEV_1_.

The “incomplete responder” and “irreversible” groups were subdivided into cumulative smoking of ≥ or <10 packs of per year.

The same phenotypes were also defined with the 2021 ATS/ERS definition of bronchial reversibility (i.e., a >10% change of FEV1 predicted value) instead of an increase of FEV1 ≥ 200 mL and ≥12%.

### 2.5. Statistics

Continuous variables were expressed as mean (±95% Confidence Interval, CI). Categorical variables were presented as number and percentage. Normal distribution was tested by using the skewness–kurtosis test. The Chi–square test and non-parametric Mann–Whitney U test were used to compare the differences between the groups. The non-parametric Kruskal–Wallis test was applied to test the difference of cumulative smoking (number of packs per year) and of the concentration of IgE against mite allergens in each age group, due to the skew distribution of these parameters. A *p* value < 0.05 was considered significant. Analyses were performed with SPSS v23.0 (IBM, Armonk, NY, USA).

## 3. Results

### 3.1. Definitions of 5 Phenotypes

Based on their response to the bronchodilator and smoking habits, the CORD patients were classified as (Figure 1):

1/full responders

2/incomplete responders, non-smoking

3/incomplete responders, smoking

4/irreversible non-smoking

5/irreversible, smoking

### 3.2. The Relative Prevalence of COPD and Asthma during the Lifespan

During the lifespan, the cumulative inhalation of noxious particles in susceptible subjects produces airways inflammation and progressive lung tissue destruction, leading to an age-related prevalence of COPD [21,22]. This cumulative smoking–age relationship was consistently found among our patients (Figure 2A,B). The relative prevalence of mite sensitization was negatively related to age (Figure 2C,D) and, since allergy is present in more than 80% of asthmatics, we hypothesized that the prevalence of asthma decreased with age.

### 3.3. The Relationship between Age and the Prevalence of the 5 Phenotypes

We showed that the prevalence of irreversible, smoking patients was positively associated with age (Figure 3A) and, consequently, this phenotype was consistent with the diagnosis of “COPD”.

A negative age-related association was observed in the full responders and the non-smoker incomplete responders (Figure 3C). This group was diagnosed as “Asthma”.

The relative prevalence of the non-smoker irreversible or smoker incomplete responders was unrelated to age (Figure 3B). This group was considered as “ACO”.

By doing so, the relative prevalence of COPD, ACO and asthma were 42, 32 and 26%, respectively. The characteristics of the so-defined asthma, ACO and COPD are compared in Table 1. Compared to asthma, COPD patients were older, almost all male, all smokers, less educated, less allergic to mites, with a more frequent history of TB, a low post-BD FEV1 and a decreased total diffusion capacity. Patients with ACO were comparable to asthmatics for gender distribution and mite allergy and to COPD for past TB; their smoking habits, age, FEV1 and diffusion capacity were intermediate between asthma and COPD.

### 3.4. The Relationship between Age and the Prevalence of ACO, in Regard to Diffusion Capacity

Our ACO patients may have been exposed to multiple environmental factors, such as exposure to tobacco smoke and indoor and outdoor pollutants, combined with allergies, all modulated by individual susceptibility. At least 2 studies have reported that COPD patients are characterized by a decreased diffusion capacity when exposed to tobacco smoke and/or biomass fumes, compared to non-exposed COPD [21,22]. Consequently, the cumulative exposure to pollutants, over one’s lifespan increases the risk of decreased diffusion. On one hand, in our ACOs, the prevalence of patients with a diffusion capacity < 80% PV was positively associated to age (Figure 3D) and considered to be ACO-like COPD. On the other hand, the prevalence of ACO with a normal diffusion capacity was negatively related to age (Figure 3E), and this group was considered to be ACO-like asthma.

The ACO-like COPD and ACO-like asthma represented 14 and 18%, respectively. The total relative prevalence was 56% (i.e., 42% + 14%) for COPD and 44% (i.e., 26% + 18%) for asthmatics. The characteristics of the asthmatic group (combining asthma and ACO-like asthma) and the COPD group (combining CPOD and ACO-like COPD) are shown in Table 2. Compared to COPD, asthmatics were more frequently female, younger, more educated, less exposed to indoor fumes, with less frequent tuberculosis history and had less severe fixed airways obstructive defects and normal total lung diffusion; they were more frequently sensitized to mites, and the majority were non-smokers.

The same analysis was obtained with bronchial reversibility defined as a >10% change of FEV1 predicted value. As shown in Figure A1 and Figure A2 (see the Appendix A), the distribution of the phenotypes, and their age/prevalence relationship, was comparable to the previous data, based on bronchial reversibility defined by a FEV1 increase of >12% and 200mL. By doing so, the prevalence of COPD, ACO and asthma were 47, 29 and 24%.

## 4. Discussion

Based on symptoms, pre- and post-BD FEV_1_/FVC and FEV_1_ and cumulative smoking, the prevalence of COPD, asthma, ACO were 42%, 26% and 32%, respectively. Based on decreased or normal diffusion capacity, there were 14% ACO-like COPD and 18% ACO-like asthma.

In Asian LMICs, asthma is a heterogeneous syndrome, due to the following main environmental risk factors: the rapid disappearance of the rural environment, increasing indoor and outdoor air pollution and a still very high prevalence of smoking (among males) [5]. Compared to Western countries, risk factors and phenotypes of COPD are also different in Asian countries, such as a high prevalence of COPD among non-smoker females [23]. Characteristics of COPD when the patient has never smoked are physiologically and radiologically different [24], as is their gender distribution. Besides the high frequency of male smoking, Asian COPD patients are characterized by frequent exposure to biomass and air pollution, post-tuberculosis bronchiectasis, low BMI, parasitosis and low use of inhalers [25].

How to treat patients with the most appropriate treatment has not yet been recommended. Indeed, drug treatment of non-smoker COPD and smokers with asthma or ACO is uncertain due to their exclusion from most clinical trials [26]. Therefore, the present study aimed to differentiate asthma, COPD and ACO in Vietnam, so that treatment based on GINA/GOLD recommendations can be applied.

First, associated with symptoms and abnormal spirometry, the exposure to noxious agents is a part of the definition of COPD in the GOLD guidelines. This exposure increases cumulatively with age and the duration of exposure. Consequently, the relative prevalence of the disease increases with age, as was observed among our smoking post-BD irreversible patients. One can argue that the FEV_1_/FVC decreases with aging, resulting in more frequent obstruction in adults > 45 years [27]. when defined as < 0.7, though obstruction was defined by LLN which is independent with age.

Secondly, we showed that mite IgE sensitivity (SPT or specific IgE) decreased with age, as repeatedly reported [28,29,30,31]. In 2011, Scichilone and colleagues reviewed the available literature assessing the impact of age on atopy: the majority of the studies (i.e., 15 among 17) reported a decline of the prevalence of atopy with aging, associated with the immunosenescence process, in both healthy subjects and individuals affected by allergic respiratory diseases [31]. More recently, in a longitudinal multinational study, over a period of 20 years, the prevalence of specific IgE sensitization to house dust mites and cats, significantly decreased as a consequence of aging [32]. Whether this age-related decrease of atopic status can diminish the prevalence of allergic asthma in the elderly population, remains questionable. Early onset of asthma is often allergic and can lead to remodeling airways while non allergic phenotypes appear later in life [32,33,34,35]. Asthma alone was reported less frequently in elderly patients, compared to younger ones [36]. Thus, we hypothesized that the prevalence of IgE-mediated asthma was negatively associated with age. In our patients, with a full reversible airways’ obstruction (i.e., the GINA definition of asthma), their relative prevalence decreased with age whatever their cumulative smoking, suggesting some smokers are not susceptible to tobacco smoke. This genetic susceptibility is sustained by data showing that the lung tissue gene-expression signature was different for ageing lungs in smoking COPD, compared to non-COPD controls [10]. Moreover, in our population, the relative prevalence among the non-smoker incomplete reversible patients was also negatively age-related. For this last group, we hypothesized that the partially fixed obstruction was due to persistent Type 2 inflammation leading to airways’ remodeling [37].

The influence of environment and aging, as risk factors for asthma and COPD have been proposed in several reviews on ACO [1,2]. The interaction of harmful effects of airborne noxious particles (increasing with age), immune inflammation (decreasing with age) and genetic susceptibility could contribute to the risk of ACO, leading to opposite relationships with age. This was observed among the non-smoker irreversible and smoker incomplete reversible patients, who were considered to be ACO patients.

These ACO patients were then compared to asthma and COPD. Compared to ACO, the asthmatics were younger, with less tobacco use, with a less frequent history of tuberculosis and less obstructive defects, while the gender distribution and atopic status were similar. Compared to ACO, the COPD were older, with a higher consumption of tobacco, more likely to be male, with more severe functional obstruction and lower atopic status, consistent with literature [1,2,3,4]. A decrease of the total lung diffusion capacity for CO was associated with COPD and ACO, compared to asthma, with a larger defect in COPD than ACO. Thus, ACO had some phenotypic characteristics of both asthma and COPD.

There are numerous review papers defining “ACO” but no consensus has been reached [1,2,3,4]. In 2015, a global expert panel discussion defined “ACO” as a persistent airflow limitation (post-BD FEV_1_/FVC < LLN) in patients older than 40 years, with cumulative smoking > 10 packs per year (or equivalent exposure to pollutants), with an history of asthma before 40 years of age (or post-BD FEV_1_ > 400 mL) and with one minor criterion (atopy and/or post-BD FEV_1_ > 200 mL and 12% and/or blood eosinophilia > 300 cells/mcL) [38]. This definition overlapped, at least partially, with the characteristics of our patients with ACO since it combined COPD (age > 40 years, exposure to smoke/pollutants, persistent airway obstruction) and asthma risk factors (asthma in the early life, atopy, post-BD FEV_1_ response). Our data are also consistent with the Dutch hypothesis, suggesting that asthma and COPD have common origins and clinical expressions and are determined both by endogenous (heredity, age, and sex) and exogenous (environment: allergens, smoking, viruses, and air pollution) factors [39]. GINA and GOLD defined ACO similarly [38]. While other authors, based on consensus of opinion, reported systems of major and minor criteria to identify ACO [40,41]. Although there were some differences between these proposal systems, they shared several key obligatory features: that patients should be 40 years of age, have persistent airflow obstruction, and a history of asthma or evidence of bronchodilator reversibility. Many of these systems do not feature environmental exposure (either cigarette smoke or biomass fuel or others). The criteria often require an FEV_1_ measurement with reversibility but also measure IgE, fractional exhaled nitric oxide, blood and sputum, eosinophilia, methacholine challenge, lung diffusion capacity or chest CT scan. These are all expensive or only limitedly available techniques in LMICs.

The Vietnamese population is exposed to numerous environmental risk factors contributing to the high relative prevalence of ACO among CORD. Although a recent large multinational study reported that ACO might be as common and more severe in LMICs than in high-income settings due to the presence of risk factors such as rapid urbanization and biomass exposure [42].

COPD is a disease “…usually caused by significant exposure to noxious particles or gases” [16]. In Western countries, COPD is more often caused by smoking habits and consequently, associated with a defect in DLCO [23]. The COPD associated to biomass exposure is also characterized by a significant decrease of DLCO, although this decrease is less pronounced when compared to tobacco smokers [43]. Thus, one feature of COPD related to environmental illness is a decrease of DLCO, although this exposure risk is not always identified. In our patients, we named ACO-like-COPD patients as those with a DLCO < 80% PV (because their relative prevalence rose with age) and ACO-like-asthma patients as those with a normal DLCO (because their relative prevalence decreased with age).

Returning to the phenotypes defining asthma and COPD, can this proposal be used in clinical practice? Based on symptoms, spirometry before and after the bronchodilator, cumulative smoking and diffusion capacity, asthma and COPD can be easily defined (Figure 4). COPD patients were smokers irreversible to BD, asthmatics were non-smoker (full or incomplete) responders or smokers full responders to BD and ACO were non-smokers irreversible to BD or smokers with incomplete response to BD [24,25]. Finally, based on decreased diffusion capacity or not, the ACO were considered ACO-like COPD or ACO-like asthma, respectively.

Our asthmatics and COPD patients should be treated on the basis of the GINA and GOLD guidelines, since the criteria of diagnosis are consistent with the guidelines. In addition to this, there are recommendations to treat patients with ACO systematically with ICS and the bronchodilator [41]. Based on the diagnosis of ACO-like asthma or ACO-like COPD, it seems logical to apply the GINA recommendations for the first group and the GOLD recommendations for the second one, instead of treating all the patients with ICSs.

## 5. Conclusions

In a Vietnamese population of Chronic Obstructive Respiratory Diseases, we defined asthma and COPD, based on clinical symptoms, cumulative smoking, spirometry with reversibility and diffusion capacity. By doing so, the prevalence of asthma and COPD were 44 and 56%, respectively, with most of the latter not requiring corticosteroids. Nevertheless, the relevance of this proposal requires testing in other populations and after patients have received specific treatments.

## Figures and Tables

**Figure 2 jpm-13-00078-f002:**
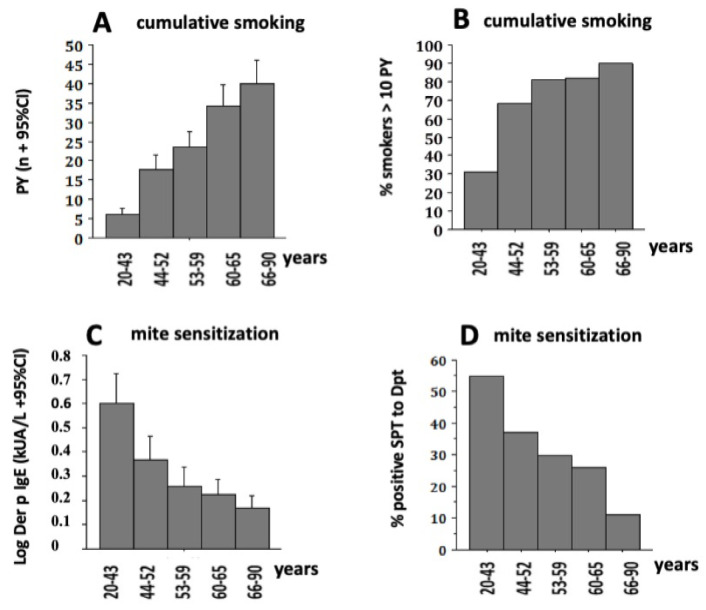
The relationship between the age groups and: (**A**) the mean cumulative smoking expressed in packs per year (among ex-smokers and current smokers, n = 443); (**B**) the relative prevalence of smokers > 10 packs per year in each age group; (**C**) the mean concentrations of Log IgE to *Dermatophagoides* (Der p) (n = 610) and (**D**) the relative prevalence of positive SPT to Dpt (n = 610). (**A**,**C**): Kruskal–Wallis test, *p* < 0.01; (**B**,**D**): Chi–square test, *p* < 0.01.

**Figure 3 jpm-13-00078-f003:**
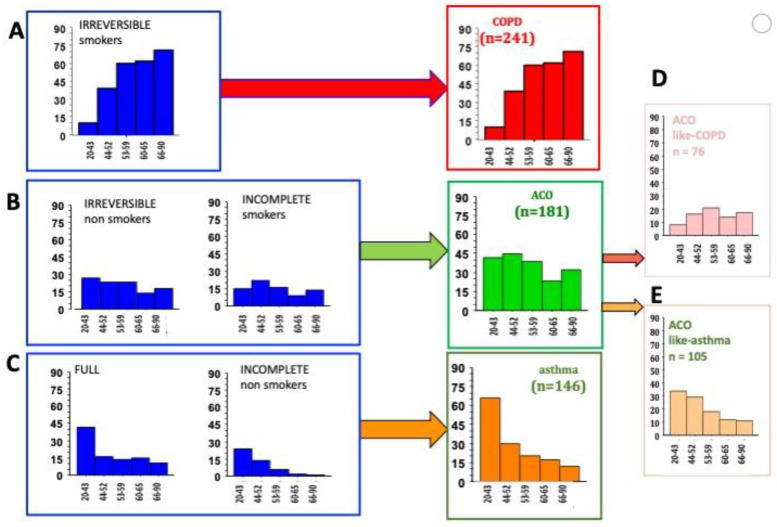
The relationship between the age groups and the prevalence of each phenotype of obstructive airway diseases. Patients were defined as (**A**) COPD, (**C**) asthma or (**B**) ACO in regard to a positive, negative or non-significant relationship with aging. (**D**): the age-related relative prevalence of ACO-like COPD (pink bar) with a DLCO < 80% PV. (**E**): the age-related relative prevalence of ACO-like asthma (light orange bar) with a DLCO ≥ 80% PV. COPD: chronic obstructive pulmonary disease. ACO: asthma—COPD overlap.

**Figure 4 jpm-13-00078-f004:**
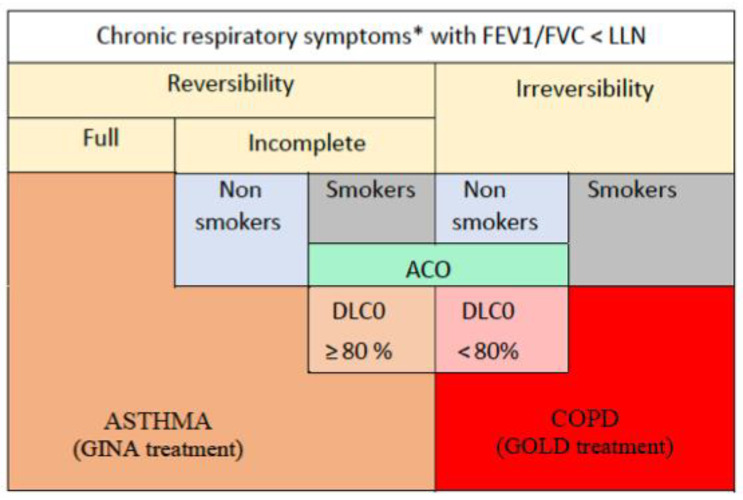
The strategy to select GINA or GOLD treatment among Vietnamese patients with chronic obstructive respiratory diseases based on symptoms, spirometry with response to BD, cumulative smoking and DLCO measurement among ACOs. GINA: Global Initiative for Asthma. GOLD: Global Initiative for Chronic Obstructive Lung Disease guidelines. COPD: chronic obstructive pulmonary disease. ACO: asthma—COPD overlap. * chronic respiratory symptoms: cough and/or sputum and/or dyspnea and/or wheezing and/or chest tightness lasting > 3 months.

**Table 1 jpm-13-00078-t001:** Characteristics of asthma, ACO and COPD (N = 568) [n (%)].

	*n* (%)	Asthma 146 (26.0)	ACO 181 (32.0)	COPD 241 (42.0)	*p*
A **Demographics and socio-economic characteristics**					
**Sex**					NS *; <0.001 ^†,‡^
Female	106 (18.7)	50 (34.2)	54 (29.8)	2 (0.8)	
Male	462 (81.3)	96 (65.8)	127 (70.2)	239 (99.2)	
**Age, years**					<0.001 *^,†,‡^
20–43	118 (20.8)	67 (45.9)	42 (23.2)	9 (3.8)	
44–52	114 (20.0)	30 (20.6)	45 (24.9)	39 (16.2)	
53–59	119 (21.0)	20 (13.7)	40 (22.1)	59 (24.9)	
60–65	109 (19.2)	21 (14.4)	26 (14.4)	62 (25.7)	
66–90	108 (19.0)	8 (5.5)	28 (15.5)	72 (29.9)	
Age, Mean (±95% CI)	54 (±1)	46 (±2)	52 (±2)	61 (±2)	<0.001 *^,†,‡^
**Level of education**				<0.025 *; 0.053 ^†^; <0.01 ^‡^	
Primary	154 (27.1)	24 (16.4)	49 (27.0)	81 (33.6)	
Secondary	162 (28.5)	40 (27.4)	54 (28.8)	68 (28.2)	
High school	145 (25.5)	42 (28.8)	41 (22.7)	62 (25.8)	
Post-secondary	107 (18.8)	40 (27.4)	37 (20.4)	30 (12.5)	
**Rural residence**	258 (45.4)	60 (41.1)	86 (47.5)	112 (46.5)	NS *^,†,‡^
B **Medical history, smoking habits and allergic sensitization**					
**History of tuberculosis (+)**	134 (23.6)	9 (6.2)	45 (24.9)	80 (33.2)	<0.001 *^,‡^; NS ^†^
**Smoking status**					<0.04 *^,†,‡^
Never	147 (25.9)	74 (50.7)	73 (40.3)	0 (0)	
Ex-smoker	245 (43.1)	39 (26.7)	63 (34.8)	143 (59.3)	
Current smoker	176 (40.0)	33 (22.6)	45 (24.9)	98 (40.7)	
**Pack-year** (mean ± 95% CI)	31 (±2)	16 (±4)	22 (±4)	39 (±3)	<0.05 *; <0.001 ^†,‡^
**Smoke ≥10** **Pack-year (+)**	352 (62.0)	36 (24.7)	75 (41.4)	241 (100.0)	<0.01 *; <0.001 ^†,‡^
**SPT to mite (+) ^a^**	159 (28.0)	59 (40.4)	59 (32.6)	41 (17.0)	NS *; <0.001 ^†,‡^
**IgE to mite (** **≥0.70 kU_A_/L) ^b^**	290 (51.1)	93 (63.7)	86 (47.5)	98 (40.7)	NS ^†^; <0.01 *^,‡^
C **Parameters of lung function [Mean (±95% CI)]**					
Post-BD FEV_1_ (%)	65 (±2)	82 (±3)	64 (±3)	55 (±2)	<0.001 *^,†,‡^
DLCO (%)	80 (±2)	95 (±2)	83 (±3)	68 (±3)	<0.001 *^,†,‡^

a: SPT to *Dermatophagoides pteronyssinus*, *farinae* and/or *Blomia tropicalis*; b: IgE to Dpt and/or Blo t and/or Der p1 and/or Der p2 and/or Der p23; CI: Confidence Interval; NS: Not significant; BD: bronchodilator; DLCO: Diffusing capacity of the Lungs for Carbon Monoxide. * Asthma versus ACO; ^†^ ACO versus COPD; ^‡^ Asthma versus COPD.

**Table 2 jpm-13-00078-t002:** Characteristics of whole asthma group (combined with ACO like-asthma) compared to the whole COPD group (combined with ACO-like COPD) [n (%)].

	All 568	Asthma 251 (44%)	COPD 317 (56%)	*p*
Male	462 (81.3)	168 (66.9)	294 (92.7)	<0.0001
**Age, years**				<0.001
20–43	118 (20.8)	101 (40.2)	17 (5.4)	
44–52	114 (20.0)	59 (23.5)	55 (17.4)	
53–59	119 (21.0)	39 (15.5)	80 (25.2)	
60–65	109 (19.2)	33 (13.1)	76 (24.0)	
66–90	108 (19.0)	19 (7.6)	89 (28.1)	
Age, Mean (±95% CI)	54.5 (±1.1)	47.4 (±1.6)	60.1 (±1.1)	<0.0001
**Level of education**				<0.01
Primary	154 (27.1)	52 (20.7)	102 (32.2)	
Secondary	162 (28.5)	65 (25.9)	97 (30.6)	
High school	145 (25.5)	72 (28.7)	73 (23.0)	
Post-secondary	107 (18.8)	62 (24.7)	45 (14.2)	
**House**				
Rural	258(45.4)	109 (43.4)	149 (47.0)	NS
Indoor fumes	476 (83.8)	191 (76.1)	285 (89.9)	<0.001
**Past tuberculosis**	134 (23.6)	22 (8.7)	112 (35.3)	<0.0001
**Smoking status**				<0.0001
Never	147 (25.9)	123 (49.0)	24 (7.6)	
Ex-smoker	245 (43.1)	74 (29.5)	171 (53.9)	
Current smoker	176 (40.0)	54 (21.5)	122 (38.5)	
**Smoke ≥10** Pack-year	352 (62.0)	71 (28.3)	281 (88.6)	<0.0001
**Pack-year (mean ± 95%CI)**	23.8 (±2.2)	8.9 (±1.9)	35.8 (±2.9)	<0.0001
**Allergy**				
**SPT to mite (+) ^a^**	159 (28.0)	98 (39.0)	61 (19.2)	<0.0001
**IgE to mite (≥0.70 kU_A_/L) ^b^**	276 (48.6)	148 (60.0)	128 (40.4)	<0.0001
**Parameters of lung function [Mean (±95% CI)]**				
Post-BD FEV_1_ (%)	65.1 (±1.7)	77.2 (±2.2)	55.4 (±2.0)	<0.001
DLCO (%)	79.8 (±1.9)	95.4 (±1.7)	67.5 (±2.2)	<0.0001

a: SPT to *Dermatophagoides pteronyssinus*, *farinae* and/or *Blomia tropicalis*; b: IgE to Dpt and/or Blo t; CI: Confidence Interval; BD: bronchodilator; DLCO: Diffusing capacity of the Lungs for Carbon Monoxide.

## Data Availability

Trial registration: NCT02517983 in clinicalTrials.gov.

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
