# Peer review of "A Proposal to Differentiate ACO, Asthma and COPD in Vietnam"

_jpm, 2022, doi:10.3390/jpm13010078_

Round 1

Reviewer 1 Report

This is a very elegant article  redacted in stylish English. Tha aim of the study is clear, the population clearly identified and randomized, with respect of Ethics Commitee.

The lung function is performed with good material and interpretation truly respects International recommendations, so that you could define easily the 5 phénotypes. Results are only descriptive but in complete accordance with the study objectives. They appears as a factual and strong evidence on text, tables and figures.

The introduction and discussion are underlying by strong citations and references. The COPD problematic in LMICs is clearly explained with part of biomass exposure, and conséquences on DLCO findings. This smart demonstration of characteriztion of COPD, asthma and ACO in Vietnam seems to be convincing to elabore  GINA/GOLD therapeutic strategy in this population.

Just a minor point in methods: could you precise in few words the contents of "The questionnaire" (line 70), which might be fifficult to understand if you don't read the referenced associated article at the same time.

Author Response

Reviewer 1

Just a minor point in methods: could you precise in few words the contents of "The questionnaire" (line 70), which might be fifficult to understand if you don't read the referenced associated article at the same time.

Response: The source and the content of the questionnaire was briefly summarized in the revised paper.

Reviewer 2 Report

The problem of appropriate diagnosis of obstructive bronchial diseases is problematic in many patients. Therefore it is still a good goal to research. However, this study presents a hypothesis without objective evidence of it. This manuscript needs important changes.

Major problems

1. Introduction emphasises AE after ICS, and it is too restrictive. Of course, it is possible to obtain adverse events, but the reader is able to recognize after reading that ICS are dangerous for asthma.

2. In methods, we have many unclear sentences 58-52. Question about randomization 1/10? Why >20 yrs but not >18 yrs?

3. for me definition of incomplete responders is not appropriate:

every good responder = FEV1>12%, and 200 ml may be: FEV1> 10% (this second criterion makes confusion) For Example Patient with severe asthma with FEV1 =60%could be also good responder if he reached 73% after BRD.

4. In results, patients with asthma, COPD and overlap syndrome should also be compared. 

5. discussion. There are too many unambiguous opinions about ageing, for example. Please check other data which show different observations on the decline of allergies in old age and the reduction of asthma. There is no critical assessment of the introduced categorization of patients and other work limitations

Author Response

Reviewer 2

Introduction emphasises AE after ICS, and it is too restrictive. Of course, it ispossible to obtain adverse events, but the reader is able to recognize after reading that ICS are dangerous for asthma.

Response: a sentence was added to point the ICS as the primary treatment of asthma

In methods, we have many unclear sentences 58-52. Question about randomization1/10? Why >20 yrs but not >18 yrs?

Response: the sentence was rewritten to clarify the selection method.

None patient < 20 years was included in this study.

for me definition of incomplete responders is not appropriate: every good responder = FEV1>12%, and 200 ml may be: FEV1> 10% (this second criterion makes confusion) For Example Patient with severe asthma with FEV1=60%could be also good responder if he reached 73% after BRD.

Response: we did not give a definition of “good responder” but only a definition of “significant responder “ (FEV1 >12% and 200 ml) as reported in the 2005 ERS-ATS interpretation standards. We agree that a severe asthmatic with FEV1 = 60% reaching 73% (and > 200ml) after BD  is a “significant responder “.

Since ERS-ATS published in 2022 new interpretation standards (ref: Stanojevic S, et al. ERS/ATS technical standard on interpretive strategies for routine lung function tests. Eur Respir J. 2022 Jul 13;60(1):2101499.) we re-analyzed our data based on a bronchodilator response of FEV1 >10 %.

In results, patients with asthma, COPD and overlap syndrome should also be compared.

Response: table 1 give detailed comparisons of the asthma – ACO and COPD.

discussion. There are too many unambiguous opinions about ageing, for example. Please check other data which show different observations on the decline of allergies in old age and the reduction of asthma. There is no critical assessment of the introduced categorization of patients and other work limitations

Response: the study of Scichilone and colleagues (reference 31) is a review paper of the available literature on the association of aging and decline of atopy, supported by 15 (among 17) studies. This reference has been reformulated in the revised paper. A recent longitudinal study, also supporting this association, was added in the discussion (ref 34). Only two studies showed a lack of decline in the prevalence of atopy with aging (cited in reference 32); moreover both studies have limitations , such as the small number of subjects.